# Deletion of *Hn**rnpk* Gene Causes Infertility in Male Mice by Disrupting Spermatogenesis

**DOI:** 10.3390/cells11081277

**Published:** 2022-04-09

**Authors:** Haixia Xu, Jiahua Guo, Wei Wu, Qiu Han, Yueru Huang, Yaling Wang, Cencen Li, Xiaofang Cheng, Pengpeng Zhang, Yongjie Xu

**Affiliations:** 1College of Life Science, Xinyang Normal University, Xinyang 464000, China; hxxu@xynu.edu.cn (H.X.); guojh@xynu.edu.cn (J.G.); ht005561@whu.edu.cn (W.W.); hanqiu96@163.com (Q.H.); huangyr@xynu.edu.cn (Y.H.); yalingw97@163.com (Y.W.); licencen@xynu.edu.cn (C.L.); chengxiaofang@xynu.edu.cn (X.C.); 2Institute for Conservation and Utilization of Agro-Bioresources in Dabie Mountain, Xinyang Normal University, Xinyang 464000, China

**Keywords:** hnRNPK, meiosis, transcriptional regulation, spermatocytes, spermatogenesis

## Abstract

HnRNPK is a heterogeneous nuclear ribonucleoprotein (hnRNP) that has been firmly implicated in transcriptional and post-transcriptional regulation. However, the molecular mechanisms by which hnRNPK orchestrates transcriptional or post-transcriptional regulation are not well understood due to early embryonic lethality in homozygous knockout mice, especially in a tissue-specific context. Strikingly, in this study, we demonstrated that hnRNPK is strongly expressed in the mouse testis and mainly localizes to the nucleus in spermatogonia, spermatocytes, and round spermatids, suggesting an important role for hnRNPK in spermatogenesis. Using a male germ cell-specific hnRNPK-depleted mouse model, we found that it is critical for testicular development and male fertility. The initiation of meiosis of following spermatogenesis was not affected in *Hnrnpk* cKO mice, while most germ cells were arrested at the pachytene stage of the meiosis and no mature sperm were detected in epididymides. The further RNA-seq analysis of *Hnrnpk* cKO mice testis revealed that the deletion of hnRNPK disturbed the expression of genes involved in male reproductive development, among which the meiosis genes were significantly affected, and *Hnrnpk* cKO spermatocytes failed to complete the meiotic prophase. Together, these results identify hnRNPK as an essential regulator of spermatogenesis and male fertility.

## 1. Introduction

Mammalian spermatogenesis is a highly complex cell differentiation process that can be divided into three precisely regulated stages: (i) spermatogonial self-renewal and differentiation; (ii) meiosis, with the genome shuffling of parental genes and production of haploid gametes; and (iii) spermiogenesis, the transformation of earl haploid round spermatids into fully differentiated spermatozoa [1,2,3]. Central to this unique biological process is meiosis, which begins in prophase of the first meiotic division that is divided into five substages: leptotene, zygotene, pachytene, diplotene, and diakinesis, and is responsible for the production of mature spermatozoa as well as for maintaining the integrity of the genome. During spermatogenesis meiosis, the chromosomes undergo significant changes that allow for the pairing of homologs and the exchange of genetic information between non-sister chromatids, and different kinds of regulatory factors are turned on or off in a specific temporal and spatial order, which is a necessary prerequisite for the normal operation of the process. Heterogeneous nuclear ribonucleoproteins (hnRNPs) are a large group of ubiquitously expressed RNA-binding proteins (RBPs) that are crucial for many aspects of cellular RNA metabolism including transcription, alternative splicing, stability, and translation, while hnRNPK is a widely studied member of the hnRNP family related to the regulation of chromosome remodeling, transcription, translation, and various signal transduction processes, and is strongly associated with the development of the nervous system and reproductive system, multiple organs, and the occurrence and development of tumors [4,5,6,7,8]. Many members of the hnRNP family have been proven to be necessary for spermatogenesis, such as hnRNPU, hnRNPL, DAZAP1, and hnRNPG-T [9,10,11,12], but there is still a lack of research on the relation between hnRNPK and spermatogenesis. The preliminary results suggest that hnRNPK may regulate spermatogenesis through its special structural characteristics, selective splicing, and expression, and there is a close relationship between them.

HnRNPK is a multifunctional protein, which mainly comprises three K homology domains (KH1, KH2, and KH3), a K-protein interactive region (KI), an N-terminal bipartite nuclear localization signal (NLS), and a nuclear shuttling domain (KNS). Mediated by these domains, hnRNPK shuttles between the nucleus and cytoplasm, binds to RNA and single or double-stranded DNA, and interacts with all kinds of proteins including transcriptional factors, splicing factors, epigenetic factors, signal molecules, and protein kinases [7]. As an important transcriptional regulator, hnRNPK has been shown to regulate many key biological processes. For example, hnRNPK recruits Pol II (RNA polymerase II) to bind to proliferation/self-renew associated genes such as *MYC*, *FGFBP1*, *CYR61*, EGFR, *PTHLH*, and cyclins to promote epidermal self-renewal [13,14]. HnRNPK can also regulate gene expression by interacting with the H3K9 methyltransferase SETDB1, the H3K27 methyltransferase EZH2, the maintenance DNA methyltransferase DNMT1, and other epigenetic factors through the KI domain [15,16,17]. These epigenetic factors all play key regulatory roles in spermatogenesis [18,19,20]. Furthermore, hnRNPK is mainly located in the nucleus of mouse spermatocytes and diffuses in abnormal spermatocytes, while the reduced expression of hnRNPK in the testis of offspring may suppress the proliferation of spermatocytes, promote apoptosis, and lead to light testicular weight and significantly impaired spermatogenesis [21,22,23]. Chapman et al. also pointed out hnRNPK and other members of the same family are involved in the regulation of spermatogenesis through selective splicing and subcellular localization [24]. In addition, mutation of *HNRNPK* is associated with Au-Kline syndrome which is described as multiple congenital malformation syndrome associated with intellectual disability [25], and the male patients had genitourinary system defects and cryptorchidism [26]. However, due to the embryonic lethality prior to 13.5 days of hnRNPK knockout mice [27], the biological functions in vivo of hnRNPK in mammalian tissue development especially about spermatogenesis remain to be expounded.

In this study, we demonstrated that the hnRNPK protein is abundantly expressed in mouse testis and presents a highly dynamic expression pattern in spermatogenesis, suggesting that hnRNPK probably also plays key roles in spermatogenesis. To understand how hnRNPK functionally influences spermatogenesis in vivo, we specifically depleted hnRNPK gene in spermatogonia by crossing *Hnrnpk*^flox/flox^ mice with *Stra8*-Cre^+/−^ transgenic mice. The loss function of hnRNPK in mouse germ cells causes male infertility, characterized by the atrophy of seminiferous tubules owing to abnormal meiosis during spermatogenesis. There was a higher apoptosis rate and a decreased expression of meiotic marker γH2A.X signals in *Hnrnpk* cKO testis. Specifically, the arrest of *Hnrnpk* cKO spermatocytes at pachytene was detected in chromosome spreads and immunostaining analysis. Meanwhile, the RNA-seq data clearly indicated that the absence of hnRNPK results in down-regulation of meiotic spermatocyte genes and depression of these genes related to transcriptional regulation, apoptotic process regulation, and chromatin binding. Our results indicate that hnRNPK modulates meiotic progression in male germ cells as an important transcriptional factor and is indispensable in spermatogenesis and male fertility in mice.

## 2. Materials and Methods

### 2.1. Ethics Statement

All animal research was approved by the institutional research ethics committee of Xinyang Normal University, and the approval number was XYEC-2021-011.

### 2.2. Generation of Hnrnpk cKO Mice

*Hnrnpk ^flox/flox^* mice were generated with the aid of Cyagen Biosciences Inc. (Suzhou, Jiangsu, China) in a C57BL/6 genetic background using TurboKnockout^®^ technology. Genotypes were identified by PCR analysis (F: 5′-GTCTCTCGCTCTGTCTTTGTGGC-3′, R: 5′- GGAAGGGCTCAGATTAAGTGGCAA-3′). The PCR products were amplified from a wild-type *Hnrnpk* allele (360 bp) or floxed allele (473 bp) or floxed heterozygote (360 bp and 473 bp). *Stra8-Cre* mice (Jax No. 008208, obtained from The Jackson Laboratory) were used to generate *Hnrnpk* conditional knockout mouse model in the present study, and *Stra8-cre* expresses improved Cre recombinase under the control of a 1.4 kb promoter region of the germ cell-specific stimulated by *Stra8* (retinoic acid gene 8) [28]. Germ cell-specific *Hnrnpk* knockout mice were constructed according to the following strategy. Briefly, 8-week-old *Stra8*-Cre males were crossed with adult *Hnrnpk ^flox/flox^* females to obtain *Stra8*-*Cre*; *Hnrnpk ^flox/+^* female mice, which were crossed with adult *Hnrnpk ^flox/flox^* males, and the *Stra8-Cre*; *Hnrnpk ^flox/flox^* male mice were identified as the conditional homozygous knockout mice (*Hnrnpk* cKO), and the littermates *Hnrnpk ^flox/flox^* male mice were used as the WT controls. Genotypes of *Stra8-Cre* mice were identified by PCR analysis (F: 5′-GTGCAAGCTGAACAACAGGA-3′, R: 5′-AGGGACACAGCATGGAGTC-3′). The PCR products amplified from a 200 bp band as Cre positive or no band as Cre negative.

### 2.3. Fertility Test

The fertility test was carried out by sexually mating 8-week-old wild-type controls and *Hnrnpk* cKO male mice with wild-type adult female mice with known fertility for at least four months. At the outset, female mice were examined daily by the appearance of a vaginal plug to determine if mating activity occurred and if reproductive behavior was normal. Cages were inspected daily to determine when pups were born and litter size was recorded

### 2.4. Real-Time Quantitative PCR (qRT-PCR) Analysis

Total RNA was isolated from testes using Trizol reagent (Takara, Dalian, Japan) according to the manufacturer’s instructions. The synthesis of first-strand cDNA was performed by reverse transcription of 1 µg total RNA using M-MLV Reverse Transcriptase (Invitrogen, Waltham, MA, USA). QRT-PCR was carried out on the Roche LightCycler 96 PCR System (Roche, Basel, Switzerland) with the following protocol: 1 cycle (50 °C for 2 min, 95 °C for 10 min) followed by 40 cycles of a two-stage temperature profile (95 °C for 15 s, 60 °C for 60 s). QRT-PCR reactions were performed using 0.4 µL of 5× diluted cDNA, 0.2 µL primers (10 nM), 10.0 µL qRT-PCR mixture (FastStart Essential DNA Green Master, Roche, Basel, Switzerland), and 9.2 µL deionized water in a reaction volume totaling 20 µL. Gene-specific primers (detailed in Appendix A) were used to determine the relative expression levels of the candidate target genes according to the standard-curve methodology, which were quantified relative to the expression of the mice *Gapdh* and *18S*
*rRNA* (housekeeping gene), by employing an optimized comparative Ct value method (ΔΔCt), and the expression level was calculated as 2^^(−ΔΔCt)^ to compare the relative expression.

### 2.5. Western Blotting Analysis

After removing the tunica albuginea of the testis, the seminiferous tubules protein samples were prepared using RIPA lysis buffer (Beyotime, Nantong, China) with the cocktail of protease inhibitor (Beyotime, Nantong, China). Protein was quantified using an Enhanced BCA Protein Assay Kit (Beyotime, China). Then, proteins (30 µg per sample) were denatured by heating at 100 °C, separated on 12.5% SDS-PAGE gels, and transferred to PVDF membranes. The blotted membranes were blocked with 5% non-fat dry milk for 1 h at room temperature, and then incubated with appropriate primary antibodies (Appendix A) overnight at 4 °C. Finally, the membranes incubated with HRP-labeled Goat Anti-Mouse IgG or HRP-labeled Goat Anti-Rabbit IgG (Beyotime) as secondary antibodies at a dilution of 1:1000 and detected by the enhanced chemiluminescence (ECL) System (Beyotime). Images were acquired and quantified using a FluorChem M imaging system (ProteinSimple, San Francisco, CA, USA).

### 2.6. Tissue Collection and Histological Examination

The testes and epididymis of the mouse were collected and weighed, fixed in 4% PFA (paraformaldehyde) solution, and then embedded in paraffin wax to cut into 5-μm-thick tissue slices using a rotary microtome. Three slices of each testis and epididymis sample were stained with H&E (hematoxylin-eosin) for histological analysis. For immunofluorescence (IF) staining, cross sections were deparaffinized with xylene (10 min) and rehydrated in a descending ethanol series (100, 90, 80, and 70% ethanol, 5 min each). Antigen retrieval was performed with 1X Tris-EDTA buffer for 10 min by keeping them in boiling, and then pretreated after being cooled to room temperature. Sections were blocked with 5% BSA for 30 min, and incubated with properly diluted primary antibodies (Appendix A) overnight at 4 °C. Then, the sections were rinsed with PBS and incubated with Alexa Fluor 488 or 555-conjugated secondary antibodies (Beyotime, A0423 or A0460, 1:200) for 2 h at room temperature and DAPI Staining Solution (Beyotime) was used to visualize the nucleus. The images were visualized using a Nikon 80i microscope with NIS-Elements software (Nikon, Tokyo, Japan).

### 2.7. Meiotic Chromosomal Spread and Immunofluorescence

Testes were collected from 4-week-old (P28) mice, and spermatocyte chromosome surface spreading was prepared as previously research [29]. Briefly, after removing the tunica albuginea, the testicular tubules were treated with the hypotonic buffer (30 mM Tris pH 8.2, 50 mM sucrose, 5 mM EDTA, 17 mM citrate dihydrate, 0.5 mM DTT, and 0.5 mM PMSF) on ice for 1 h, and then suspended in 100 mM sucrose buffer (pH 8.2) and gently dispersed to form single cells suspension. Subsequently, the cell suspension was smeared onto adhesive slides and fixed with an equal volume of fixative buffer (1% PFA, 0.15% Triton X-100, 10 mM sodium borate, pH 9.2). Finally, after drying overnight in a humidity box, the slides were washed with PBS (pH 7.4) three times and blocked with 5% BSA for 30 min before incubation with immunostaining with anti-SCP3 and anti-hnRNPK antibody (Appendix A) according to the above immunofluorescence protocol.

### 2.8. TUNEL Assay

The TUNEL reaction was performed following the manufacturer’s introduction by using the riboAPO One-Step TUNEL Apoptosis Kit (RiboBio, C11012-1, Guangzhou, China). In brief, the 5-μm thick sections after deparaffinization and rehydration were treated with proteinase K (20 μg/mL) for 15 min and washed in PBS for 5 min. The sections were then applied and kept on 1x TdT enzyme equilibration buffer for 5–10 min at room temperature. Subsequently, the samples were incubated with the TdT enzyme reaction mixture in a humidified chamber at 37 °C for 2 h. After stopping the reaction with 2x SSC buffer at room temperature for 15 min and washing 3 times in PBS, the sections were stained with DAPI for 30 min at room temperature and were washed in PBS 3 times again. The images were captured using a Nikon 80i microscope system.

### 2.9. RNA-Seq Library Preparation and Analysis

RNA was isolated from 3-week-old (P21) and 4-week-old (P28) mouse testis using RNAiso Plus reagent (Takara). The qualified total RNA was further cleaned up by using RNAClean XP Kit (Beckman, Krefeld, Germany) and RNase-Free DNase Set (QIAGEN, Hilden, Germany). After passing through the quality inspection, the RNA-Seq library was constructed using the VAHTS Stranded mRNA-seq Library Prep Kit for Illumina^®^ (Vazyme, NR602-02, Nanjing, China), and sequenced on the Illumina Hiseq 2000/2500 NextSeq with paired-end sequencing in Shanghai Bohao Biotechnology Co., Ltd. After prefiltering the raw data by removing the joint sequence and low-quality reads, the preprocessed reads were conducted to align the mouse genome GRCm38.p4 (mm10) using Hisat2 estimated by (version: 2.0.4, Dallas, TX, USA) alignment software for genome mapping. The expression level of each gene was normalized by FPKM (fragments per kilobase of exon model per million mapped reads) using Stringtie software (version 1.3.0, Baltimore, MD, USA). Genes exhibiting a log_2_Fold Change ≥ 1.0 and Q-value ≤ 0.05 were considered to be significantly differentially expressed in the testes derived from *Hnrnpk* cKO group compared to in WT control groups. The biological functions of identified differentially expressed genes were annotated using gene ontology analysis and the KEGG pathway analysis.

### 2.10. Statistical Analysis

All the experiments were carried out at least in triplicate and the values were shown as mean ± SEM. Statistical significances were measured based on *t*-test (two groups) or one-way ANOVA using the SPSS17.0 software. *p*-value ≤ 0.05 (*) was considered as significant difference, and *p*-value ≤ 0.01 (**) was considered as a very significant difference between the two groups.

## 3. Results

### 3.1. HnRNPK Is Preferentially Expressed in the Mouse Testes

We screened ten organ samples by qRT-PCR quantitative analysis. *Hnrnpk* mRNA expression levels were higher in the testis and brain, which were significantly higher than in the other tissues including heart, liver, spleen, lung, kidney, iWAT, BAT, and muscle (*p* < 0.05) (Figure 1A). During postnatal testicular development, the mRNA levels of *Hnrnpk* decreased gradually from postnatal day 0 (P0) to postnatal day 14 (P14), when spermatocytes progress into the late pachytene stage, and then increased drastically from P14 onward (Figure 1B), coinciding with the first appearance of pachytene spermatocytes in the cycle of the seminiferous epithelium. This expression profile is in agreement with what has been previously described in rats [30]. The dynamic expression patterns suggest that the RNA binding protein hnRNPK may be differentially regulated in different germ cell types and may imply its crucial and potential roles during testicular development and spermatogenesis.

To further define hnRNPK subcellular localization during spermatogenesis, we carried out immunofluorescence tissue staining using hnRNPK and γ-H2A.X antibodies to detect it on the testicular paraffin section in adult testes. We observed the expression of hnRNPK throughout most of the male germ cell development, including the mitotic spermatogonia, meiotic spermatocytes (pre-leptotene to diplotene), metaphase stages spermatocytes, round spermatids, and early elongating spermatids (Figure 1C). Interestingly, strong hnRNPK immunoreactivity was detected in undifferentiated spermatogonia, pachytene and diplotene spermatocytes, metaphase I and II, round spermatids, and early elongating spermatids (steps 9–10), whereas weak hnRNPK staining was in differentiated spermatogonia, preleptotene, leptotene, and zygotene spermatocytes, and no staining in late elongated spermatids (steps 11–16) (Figure 1D,E). It is noteworthy that hnRNPK was also dynamically expressed in pachytene spermatocytes at different developmental stages. In general, hnRNPK showed weak expression in early pachytene spermatocytes (stage I–III), displaying an increased trend throughout development. In addition, hnRNPK protein was predominantly observed in the nucleus of primary spermatocytes, while the proportion of hnRNPK expression in the cytoplasm increased gradually in metaphase I and II. Together, these data indicate that hnRNPK is highly expressed in the nuclei of spermatogonia, pachytene spermatocytes, and diplotene spermatocytes, and the cytoplasm of metaphase stages spermatocytes and may play crucial roles in spermatogenesis.

### 3.2. Generation of Germ Cell-Specific Hnrnpk Conditional Knockout Mice

Biallelic knockout of *Hnrnpk* results in mouse embryonic lethality, while haploinsufficiency causes developmental defects [27], precluding further research of the roles of *Hnrnpk* in postnatal testes development and spermatogenesis. To overcome this obstacle, we generated an *Hnrnpk* loxP mouse model, which allowed us to delete *Hnrnpk* in a cell- specific manner using the Cre-loxP approach. The full-length *Hnrnpk* gene (GenBank: NM_001301341.1) contains 18 exons, while the initiation codon coding sequence ATG is located at the 4th exon (Figure 2A), implying that this coding region could be indispensable for proper *Hnrnpk* functions in vivo. Therefore, we decided to generate *Hnrnpk* floxed allele (*Hnrnpk*
^*flox/flox*^) by inserting two parallel loxP sites, one in intron 3 and the other one in intron 7. In doing so, the initiation codon region of *Hnrnpk* would be deleted in the progeny after Cre-mediated recombination in the targeted cell types. The *Hnrnpk ^flox/flox^* mice are viable and healthy, demonstrating the *Hnrnpk* flox allele that we generated is available for the following experiment. Next, by crossing *Hnrnpk ^flox/flox^* mice with *Stra8-Cre* mice, we generated the male *Stra8*-*Cre*; *Hnrnpk ^flox/flox^* (*Hnrnpk* cKO) mice (Figure 2B), in which *Hnrnpk* is inactivated specifically in early-stage spermatogonia beginning at P3 testes (postnatal day 3) [28,31]. PCR genotyping was used to distinguish WT and loxP alleles (*Hnrnpk ^flox/flox^* homozygote and *Hnrnpk ^flox/+^* heterozygote) (Figure 2C). Further, PCR amplification using cross knock primers and DNA sequencing (Appendix A) confirmed that the deleted zone containing entire exon 4, 5, 6, and 7, was 284 bp compared to the mRNA sequence (GenBank: NM_001301341.1), and was 3503 bp containing partial intron 3, 7, and entire intron 4, 5, 6, exon 4, 5, 6, and 7, compared to the DNA sequence (GenBank: NC_000079.6). The recombined *Hnrnpk* allele was truly null because of deleted initiation codon ATG and 284 bp coding sequence (Appendix A). Thus, we successfully generated *Hnrnpk* cKO mice.

### 3.3. HnRNPK Is Critical for Male Germ Cell Development and Male Fertility

*Hnrnpk* cKO male mice are viable and developed normally, with no appearance and activity defects (Figure 3A). However, their testes were smaller than the WT littermate control (Figure 3B), and the testes weight of the *Hnrnpk* cKO mice at the age of P56 were strikingly reduced compared with the littermates control (WT control: 98.83 ± 6.30 mg; n = 3; *Hnrnpk* cKO: 34.87 ± 1.80 mg; n = 3). Both caput and cauda epididymides of the *Hnrnpk* cKO male mice were also smaller compared to those of their age-matched control (Figure 3B). To explore at which stage the testes exhibited decreased size and reduced weight in the *Hnrnpk* cKO mice, we collected P5, P7, P14, P21, P28, P42, P56, and P84 control and *Hnrnpk* cKO mouse testes, and examined the ratio of testes weight and body weight to eliminate the weight errors. The results showed that the ratio of testes weight and body weight of the *Hnrnpk* cKO male mice was sharply decreased compared with the littermates control starting at P14 till to adult (*p* < 0.01) (Figure 3C).

Further, to explore what matters were involved in testes weight decreasing, we first examined the existence of the germ cells and somatic cells by H&E staining. During the first wave of murine spermatogenesis, no major differences in testicular histology were seen between the littermates control and *Hnrnpk* cKO testes in younger than P7 mice. However, starting at the age of P14, the seminiferous tubules of *Hnrnpk* cKO testes are narrower in diameter compared to the littermates control testes, and contained many vacuoles, indicative of abundant germ cell depletion, which corresponded to the depletion of spermatids in the epididymis of the *Hnrnpk* cKO males (Figure 3D). Additionally, to confirm the effects of hnRNPK deficiency on male fertility, we performed a breeding experiment by mating WT control or *Hnrnpk* cKO male mice with WT females of tested fertility for 4 months. The continuous breeding observation demonstrated that *Hnrnpk* cKO males were infertile (Table 1). In addition, the expression of hnRNPK was further detected by Western blot and tissue immunofluorescence using P28 testes, and the result indicated that the expression level of hnRNPK protein was significantly reduced in *Hnrnpk* cKO mice (Figure 3E,F). The limited expression of hnRNPK protein present in the testes of *Hnrnpk* cKO mice probably comes from Sertoli cells and Leydig cells or other somatic cells in testes. In brief, our results demonstrate that hnRNPK has a key role in spermatogenesis, which is critical for male germ cell development and male fertility.

### 3.4. Ablation of hnRNPK in Postnatal, Premeiotic Male Germ Cells Was Unable to Proliferate Normally and Apoptosis Was Increased

The second week of male mouse postnatal development is marked by an important proliferation of germ cells along with an increase in testis volume. However, starting at the age of P14, the seminiferous tubules of *Hnrnpk* cKO testes contained numerous vacuoles, obviously a smaller number of testes cells compared to the littermates control testes (*p* < 0.05) (Appendix A). The lumen diameter of epididymal tissue in adult *Hnrnpk* cKO mice was not significantly different from that in the WT control group, but there were no clusters of sperm in the epididymal lumen in *Hnrnpk* cKO mice (Appendix A). Then, we further sought to identify and investigate the cell types of spermatogenesis affected by the loss of *Hnrnpk*. In the *Hnrnpk* cKO mice, the proportion of SOX9^+^ Sertoli cells at P14 (Figure 4A,F), and PLZF^+^ undifferentiated spermatogonia at P7 (Figure 4B,G) were comparable with those of the littermates control, indicating no discernible defects in Sertoli and spermatogonia (*p* > 0.05). However, there was a significant reduction of γH2A.X^+^ meiotic cells (a marker of meiotic DNA damage response) in the *Hnrnpk* cKO testes (*p* < 0.05) (Figure 4C,H). Furthermore, most control seminiferous tubules were packed with γH2A.X labeling in the XY body with dotted staining; only sparsely populated spermatocytes were observed in *Hnrnpk* cKO testes at this stage (Figure 4G). Normally, along with the progression of meiosis, γH2A.X will restrict to the XY body at the pachytene and diplotene stage. These results indicate that a large population of *Hnrnpk* cKO spermatocytes undergoes progressive depletion during the meiotic prophase.

To investigate what caused the spermatocytes depletion in *Hnrnpk* cKO male mice at this stage, we first detected proliferation of the germ cells in P14 mouse testis using Ki67 staining, a proliferative cell marker that labels cells in the G2 and M-phase of the cell cycle. We found that the frequency of tubules containing germ cells immunostaining of Ki67 in the *Hnrnpk* cKO were significantly less than that of the WT control group at P14 (Figure 4D,I). In addition, the TUNEL assay also demonstrated that the apoptosis signal was dramatically more in the *Hnrnpk* cKO mice testes compared with the WT control littermates (Figure 4E,J). These results indicate that the decrease of spermatocytes in *Hnrnpk* cKO mice testes is probably involved in the inhibited proliferation of germ cells, and apoptosis also had an effect.

### 3.5. HnRNPK Is Required for Meiotic Prophase Completion in Male Spermatocytes

In order to further explore the stage of the primary defect appeared in the *Hnrnpk* deficient spermatocytes, we then investigated hnRNPK along with γH2A.X staining at the crucial transformation stage of spermatocytes. In the control spermatocytes, zygotene and diplotene spermatocytes were detected by co-immunostaining of hnRNPK and γH2A.X in the stage XI seminiferous tubules (Figure 5A). In contrast, the seminiferous tubule stage XI of *Hnrnpk* cKO that contain γH2A.X^+^/hnRNPK^-^ zygotene spermatocytes were observed, while no γH2A.X^+^/hnRNPK^-^ diplotene spermatocytes were detected. Meanwhile, the pachytene spermatocytes were γH2A.X^+^/hnRNPK^-^ in the stages V and X of *Hnrnpk* cKO were detected, suggesting that *Hnrnpk* cKO spermatocytes can reach late pachytene but fail to reach diplotene. Then, the arrest of *Hnrnpk* cKO spermatocytes at pachytene was further confirmed by chromosome spreads with immunofluorescent double staining of hnRNPK and SYCP3 (the meiotic chromosome axial element marker) (Figure 5B). Among all SYCP3-positive primary spermatocytes, the WT control spermatocytes exhibited all stages of prophase I (from zygotene to diakinesis), and hnRNPK was found mainly in chromosome axes of spermatocyte and its levels increased from zygotene to pachytene. However, the most advanced developing step observed in *Hnrnpk* cKO chromosome spreads was to the late pachytene stage, and hnRNPK negative diplotene and diakinesis spermatocytes were not detected, demonstrating the incapacity of meiosis chromosomes to undergo condensation and progress from pachytene to diplotene in the deficiency of hnRNPK. These observations indicated that *Hnrnpk* cKO spermatocytes failed to complete the meiotic prophase, and hnRNPK is essential for the completion of the meiotic prophase and spermiogenesis.

### 3.6. Transcriptome Analyses of Hnrnpk cKO Testes

P21 and P28 represent respectively a developmental stage along with the appearance of round spermatid and spermatozoa during the first wave of spermatogenesis [32] and with the significant changes in the distribution of germ cell populations, which were observed at the two time points between *Hnrnpk* cKO and WT control mice. To investigate gene expression changes underlying the spermatogenic defects caused by *Hnrnpk* deletion, we performed RNA-sequencing (RNA-seq) on P21 and P28 testes from WT control and *Hnrnpk* cKO mice. Principal component analysis (PCA) showed that the overall transcriptomes of the P21 WT and *Hnrnpk* cKO, P28 WT and *Hnrnpk* cKO groups were separated (Figure 6A). The PCA projections revealed that the transcriptomes of P21 *Hnrnpk* cKO and P28 hnRNPK cKO testes were different from those of the P21 WT and P28 WT group, respectively. In P21, we found 1101 and 2045 genes significantly up- and down-regulated (*Q* < 0.05, log_2_fold change ≥ 1), respectively, in *Hnrnpk* cKO mice compared to controls (Figure 6B and Appendix A). In P28, 5555 and 1758 genes were significantly up- and down-regulated, respectively (Figure 6C and Appendix A). Both up-regulated and down-regulated genes exhibited considerable overlap, with the 963 up-regulated genes and 795 down-regulated genes showing overlap, respectively (Figure 6D,E, Appendix A). These results indicate that, as an important transcription factor, hnRNPK can affect gene expression both positively and negatively in testis development.

Using gene ontology (GO) enrichment analysis, the overlapped down-regulated genes were mainly enriched in the spermatogenesis, flagellated sperm motility, acrosomal vesicle, spermatid development, and male gonad development (Figure 6G). The up-regulated genes in *Hnrnpk* cKO testis were enriched in the transcriptional regulation, cell migration, protein phosphorylation regulation, apoptotic process regulation, cell junction, chromatin binding, and lipid metabolic process not related to spermatogenesis signaling pathways (Figure 6F). To further analyze hnRNPK-regulated germ cell expression genes, we compared the overlapped differentially expressed genes with the previously published single-cell RNA-seq data of spermatogenic cells [33]. The reanalysis of the data demonstrated that the up-regulated genes in *Hnrnpk* cKO were less expressed in spermatogenic cells, only 22 genes were found in the C1 group which contained all of the mitotic spermatogonial cells (from A1 to B type) and preleptotene spermatocytes (from G1 to early S phase) (Figure 7). In contrast, most of the down-regulated genes in *Hnrnpk* cKO (370 genes) were overall lower expressed in spermatogonia and higher expressed in prophase spermatocytes, including cluster C2 (middle S and late S phase preleptotene spermatocytes), cluster C3 (leptotene and zygotene spermatocytes), cluster C4 group (pachytene spermatocytes), and cluster C5 (diplotene spermatocytes, MI and MII spermatocytes, as well as steps 1–2 spermatids). Among down-regulated genes, 84.6% of the genes were in clusters C4 and C5, including genes expression peak in pachytenes and diplotenes. Most of these genes code for meiosis-relating proteins such as *Spink2* (a strong inhibitor of acrosin) and *Ccna1* (a regulatory subunit of the cyclin-Cdk complex), and the absence of the functional proteins induces male sterile due to a blockage of meiosis [3,34,35,36]. This demonstrates significant disturbances in the spermatogenesis in the deficiency of hnRNPK, which also implied the vital role of hnRNPK in the process of spermatogenesis. Collectively, the above data revealed that the deficiency of germ cell-specific hnRNPK disrupted the expression of genes involved in multiple biological processes, among which the genes related to the meiosis process were sharply affected, and *Hnrnpk* cKO spermatocytes failed to complete the meiotic prophase.

To validate the RNA-seq results, 30 representative genes with the different expression profiles were selected according to RNA-Seq data, and their expression levels calculated via qRT-PCR in triplicates (three biological replicas for each testis sample). In particular, we chose 16 down-regulated genes that upregulated in the pachytenes and diplotene spermatocytes population (*Tssk4*, *Tssk5*, *Spata4*, *Spata9*, *Spata16*, *Sun5*, *Prss21*, *Pgam2*, *Adad1*, *Ccna1*, *Spaca1*, *Drc7*, *Ccdc113*, *Catsper1*, *Catsper2*, and *Spink2*), and 14 up-regulated genes that reflected the Leydig and Sertoli cell population (*Amhr2*, *Clu*, *Sox9*, and *Gas6*), transcription regulation (*Foxo3*, *Hdac6*, and *Dnmt3a*), and protein phosphorylation (*Jak1*, *Akt1*, *Sparc*, *Gstm1*, *Fasn*, *Ccnd2*, and *Fkbp5*). The dynamic expression patterns of all the selected genes in the four groups were in agreement with both RNA-Seq analyses (Appendix A). Additionally, there was a high correlation between RNA-seq and qRT-PCR data (Pearson R are 0.88 and 0.94 for P21 and P28, respectively), thus supporting the validity of our RNA-Seq data.

## 4. Discussion

Although previous studies have reported that hnRNPK is dynamically expressed during mammal spermatogenesis and might be functional partners for regulation of RNA processes during spermatogenesis [22,30,37], the precise function in spermatogenesis remains unclear. In this study, we explored the role of hnRNPK in the spermatogenesis by generating germ cell-specific *Hnrnpk* cKO mice using *Stra8*-Cre, which is taken to be expressed later in spermatogonia cells on P3 [28]. The important function of hnRNPK in spermatogenesis was strongly suggested by the dynamic expression and up-regulation of hnRNPK expression in the pachytene and diplotene spermatocytes in testes sections (Figure 1). Furthermore, *Hnrnpk* cKO male mice were infertile with about 50% decreases in the testis size of adults and displayed a distinct meiotic arrest phenotype. Transcriptomic analysis indicated that the germ cell-specific deficiency of hnRNPK caused the down-regulation of several miosis-related genes, supporting that hnRNPK might play a crucial role in meiotic regulation during spermatogenesis.

The detailed expression pattern of hnRNPK in male germ cells was validated using immunofluorescence histochemistry. The results showed that protein hnRNPK was expressed in spermatogonia, spermatocytes, round spermatids, and early elongating spermatids, and no expression was seen in mature sperm. The present findings are consistent with previous research that hnRNPK protein is abundantly expressed in later spermatogenesis as shown by in iTraq proteome analysis and immunohistochemical examination in mice or rats [22,24,30]. However, our results are inconsistent with research on pigs [38], which demonstrates that hnRNPK protein is expressed in mature sperm cells. This discrepancy may be due to the variations between species. In addition, it is also worth noting that hnRNPK protein was translocated from the nucleus to the cytoplasm of spermatocytes at meiotic-to-postmeiotic transition. The dynamics of nuclear-cytoplasm translocation of hnRNPK in germ cells suggested that the function of hnRNPK is different in spermatogenesis. In pachytene and diplotene spermatocytes, the hnRNPK expression level is relatively high and mainly expressed in the nucleus, indicating that hnRNPK is more involved in transcriptional regulation, and the transcriptional activity is the highest during meiosis at these two stages [32,33]. Another intriguing expression change is that hnRNPK undergo relocalization from the nucleus to the cytoplasm in post-meiotic transition spermatocytes, maybe coincident with their functional requirement, suggesting that hnRNPK is more involved in post-transcriptional regulation during this period and the timing of hnRNPK activation is critical for the metaphase/anaphase transition and normal spermatids development and function.

HnRNPK is a multifunctional protein, not only has the function of regulating RNA, but also can bind DNA directly or indirectly to participate in transcription regulation and chromatin remodeling. As a transcription factor, hnRNPK could directly bind to promoter regions or interact with the Pol II complex and associate transcription factors to modulate gene transcription [13,15,27,39]. In addition, hnRNPK interacts with many lncRNAs (long noncoding RNAs) to regulate lncRNAs nuclear enrichment and lncRNA-mediated transcription, participating in the self-renewal of naive embryonic stem cells (nESCs), somatic reprogramming, tumorigenesis, neural differentiation, and X-chromosome inactivation [40,41,42]. In this study, *Hnrnpk* cKO male mice were infertile and severely disturbed the first wave of spermatogenesis. The number of spermatogonia and Sertoli cells had no change in P14 mice, while the number of primary spermatocytes was prominently lower in hnRNPK deficiency mice than the WT control. It seemed that hnRNPK protein functioned mainly at the primary spermatocyte stage. Furthermore, the increased number of apoptotic spermatocytes and reduced proliferative activity of spermatocytes implied that hnRNPK protein played a dominant role in spermatocyte survival. In addition, our chromosome spread results also demonstrated that hnRNPK was essential for spermatocytes during meiotic prophase I, and the distribution of hnRNPK was dotted along the autosomal in pachytene and diplotene spermatocytes with higher expression level. It should also be noticed that hnRNPK was dominantly located in the nucleus of spermatocytes, while pachytene and diplotene spermatocytes are the two most active transcriptional stages in meiosis. According to research, hnRNPK also interacts with SETDB1, an important epigenetic factor regulating meiosis, to mediate H3K9me3 modification in spermatogenesis [19,43,44]. Therefore, we believe that hnRNPK may mainly participate in the regulation of spermatogenesis as a transcription factor in meiosis. The following RNA-Seq data indicated that the absence of hnRNPK results in down-regulation of spermatocyte meiotic genes and depression of these genes related to transcriptional regulation, apoptotic process regulation, and chromatin binding, which could explain, to some extent, the phenotypic characteristics of *Hnrnpk* cKO mice on molecular levels, suggesting the failure to accurately complete meiosis and to turn into postmeiotic spermatid development stages. The further analysis of hnRNPK-regulated germ cell expression genes with the previously published single-cell RNA-seq data of spermatogenic cells suggest that hnRNPK can be both a coactivator and corepressor in meiosis by regulating the expression of a series of downstream important genes. These regulated genes include spermatogonial differentiation genes (such as *Spink2*, *Pgam2*, *Tssk4*, *Tssk5*, *Spaca1*, *Foxo3*, *Hdac6*, and *Dnmt3a*). It is not known whether hnRNPK directly regulates the expression of these genes or whether hnRNPK indirectly affects their expression by affecting other direct downstream genes. However, this needs further research in the future. It would therefore be useful to reveal the function of hnRNPK in the regulatory network of spermatogenesis using hnRNPK Chip-seq analysis for spermatocytes and round spermatids, respectively.

## 5. Conclusions

In summary, we found that hnRNPK is dynamically expressed in male germ cells and may play a key role in spermatogenesis through transcription regulation. Deficiency of hnRNPK leads to abnormal spermatogenesis and most of the germ cells are lost in adult testes. Further results indicate that hnRNPK is probably essential for germ cell meiosis and the death of hnRNPK-deficiency germ cells likely results from the defects of meiosis. Our findings provide novel insights into the molecular basis underlying male sterility and new ideas for the research and application of male reproductive health.

## Figures and Tables

**Figure 1 cells-11-01277-f001:**
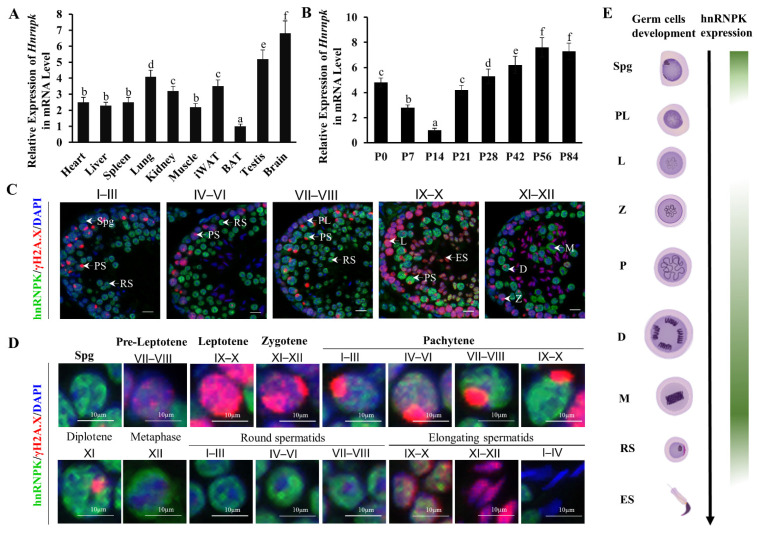
HnRNPK shows a dynamic expression pattern during spermatogenesis. (**A**) *Hnrnpk* expression in different tissues of mice. BAT, brown adipose tissue; iWAT, inguinal white adipose tissue. The difference between different letters is significant (*p* < 0.05), while the difference between the same letters is not significant (*p* > 0.05); n = 3. (**B**) The expression of *Hnrnpk* in testicular tissues of mice at different development stages. The difference between different letters is significant (*p* < 0.05), while the difference between the same letters is not significant (*p* > 0.05); n = 3. (**C**) Representative images of hnRNPK and γH2A.X immunostaining on WT adult testicular cross-sections in different stages of seminiferous tubules. (**D**) Immunostaining of hnRNPK and γH2A.X on different types of spermatogenic cells. Spg, spermatogonia; PL, preleptotene spermatocytes; L, leptotene spermatocytes; Z, zygotene spermatocytes; PS, pachytene spermatocytes; D, diploid spermatocytes; M, meiosis I metaphase spermatocyte; RS, round spermatids; ES, elongated spermatids; I-XII, the stages of spermatogenic epithelial cycle; bar = 10 μm. (**E**) Schematic diagram of hnRNPK expression in male germ cells.

**Figure 2 cells-11-01277-f002:**
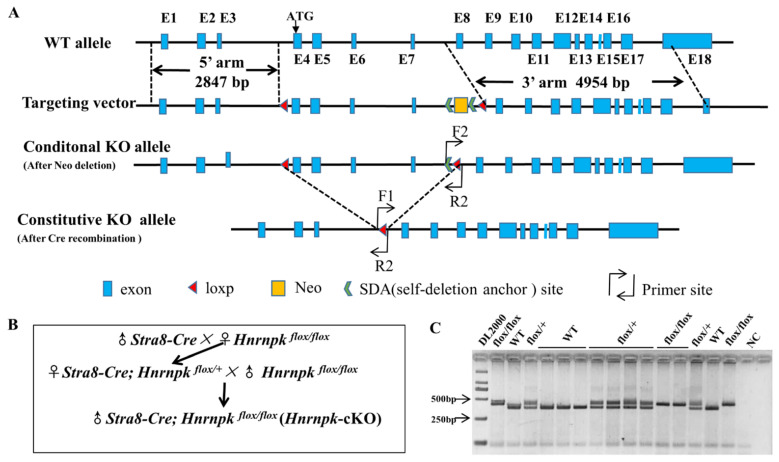
Design, mating strategy, and genotyping of *Hnrnpk* knockout targets in mice. (**A**) Schematic drawing of the targeting strategy for the generation of *Hnrnpk^flox/flox^* and *Hnrnpk germ* cell-specific knockout (*Hnrnpk* cKO) mice. Exons 4, 5, 6, and 7 were flanked by loxP and will be removed by Cre recombinase. The locations of the primers used for genotyping are shown as bent arrows. (**B**) Mating strategies of *Hnrnpk* cKO mice. (**C**) Identification of *Hnrnpk* cKO mice.

**Figure 3 cells-11-01277-f003:**
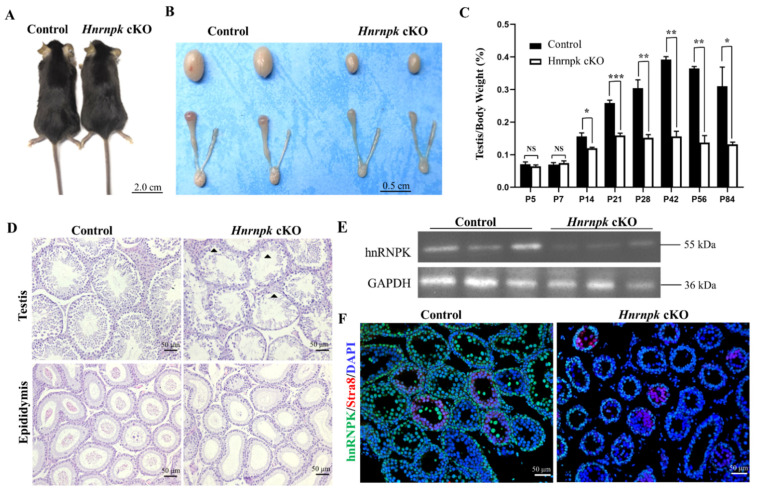
HnRNPK is essential for male fertility. (**A**) No significant difference was found in appearance and growth between WT control (26.4 ± 1.8 g) and *Hnrnpk* cKO mice (26.1 ± 1.7 g) (*p* > 0.05); n = 6. The images show mice from WT control and *Hnrnpk* cKO at P56. (**B**) Representative image of testes from WT control and *Hnrnpk* cKO mice at P56. (**C**) Testis weight-to-body weight ratio of WT control and *Hnrnpk* cKO mice of P5, 7, 14, 21, 28, 42, 56, and 84. At least three mice per genotype per time point were used for the analysis. Data are show as the mean ± SEM; n = 6. *p* < 0.001 (***), *p* < 0.01 (**), *p* < 0.05 (*), no significant difference (NS). (**D**) Histology of the testes and cauda epididymides from WT control and *Hnrnpk* cKO mice at P56. (**E**) Immunoblotting detecting the expression of protein hnRNPK in WT control and *Hnrnpk* cKO mice testes. The expression level of GAPDH was used as an internal control. (**F**) Immunofluorescence detecting the expression of protein hnRNPK in premeiotic germ cells in WT control and *Hnrnpk* cKO mice testes.

**Figure 4 cells-11-01277-f004:**
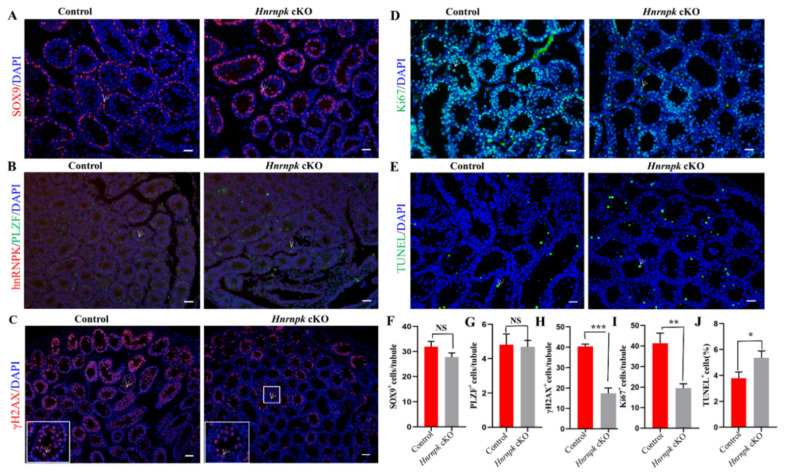
Immunofluorescence detection of different cell types in WT control and *Hnrnpk* cKO mice testes. (**A**) P14 WT control and *Hnrnpk* cKO testicular tissue sections with γH2A.X staining. The white arrow shows pachytene spermatocytes with γH2A.X. (**B**) 7-day-old WT control and *Hnrnpk* cKO testicular tissue section hnRNPK and PLZF staining. The white arrow shows PLZF staining cells. (**C**) P14 WT control and *Hnrnpk* cKO testicular tissue sections Sox9 staining. The white arrow shows Sox9 staining cells. (**D**) The proliferation of WT control and *Hnrnpk* cKO seminiferous tubules at P14 was analyzed with the Ki67 staining. The white arrow shows Ki67 staining cells. (**E**) Apoptosis of WT control and *Hnrnpk* cKO seminiferous tubules at P14 was analyzed with the TUNEL assay. The green were TUNEL-positive germ cells. (**F**–**J**) The number of γH2A.X, PLZF, Sox9, Ki67, and TUNEL-positive germ cells per seminiferous tubules, respectively. At least 200 tubules were counted for each testis (n = 3). Data are shown as the mean ± SEM. *p* < 0.001 (***), *p* < 0.01 (**), *p* < 0.05 (*), no significant difference (NS). Bar = 50 μm.

**Figure 5 cells-11-01277-f005:**
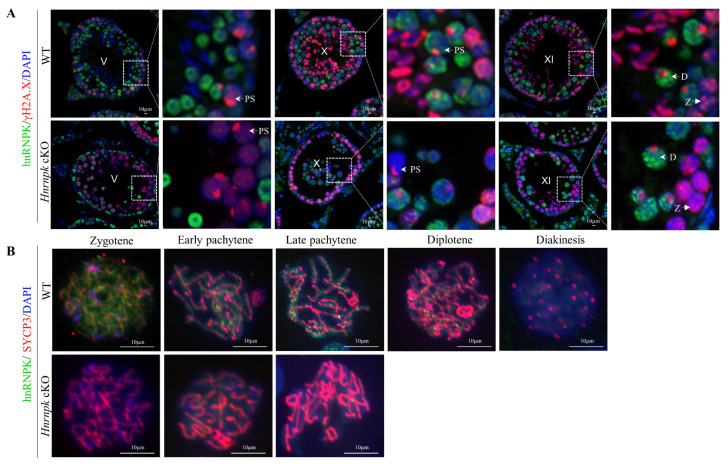
Immunofluorescence detection of different types of germ cells. (**A**) Immunofluorescence detection of hnRNPK and γH2A.X in testicular tissue from P42 mice. PS, pachytene spermatocyte; D, diploid spermatocyte; Z, zygotene spermatocytes; V, X, XI, the stages of spermatogenic epithelial cycle. (**B**) Chromosome spreading combined with immunofluorescence detection of hnRNPK and SYCP3 expression and localization. Bar = 10 μm.

**Figure 6 cells-11-01277-f006:**
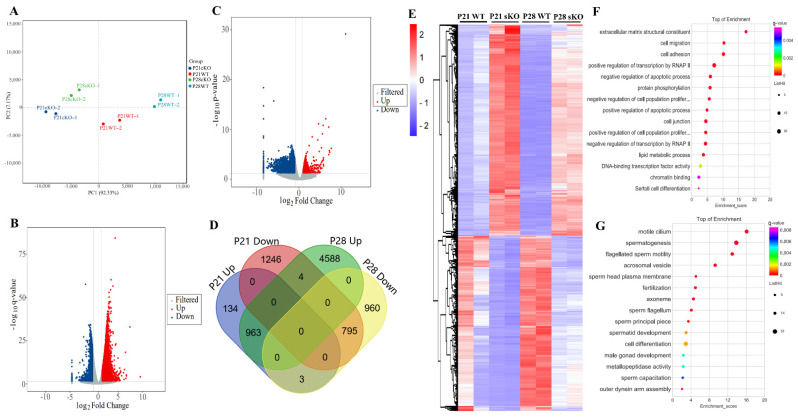
HnRNPK is involved in transcriptional regulation during spermatogenesis. (**A**) PCA analysis of RNA-Seq in each group (n = 2). (**B**,**C**) Volcano plot of the differentially expressed genes in *Hnrnpk* cKO testes compared with the WT control at P21 and P28, respectively. Blue dots and red dots represent the significantly down-regulated and up-regulated genes, respectively (Q value < 0.05, fold change of RPKM > 2), and gray dots represent unchanged genes. (**D**) Intersection Venn diagram of differentially expressed genes in P21 and P28. (**E**) The heatmap of overlapping differentially expressed genes between P21 and P28. (**F**) GO analysis of overlapping up-regulated genes between P21 and P28. (**G**) GO analysis of overlapping down-regulated genes between P21 and P28.

**Figure 7 cells-11-01277-f007:**
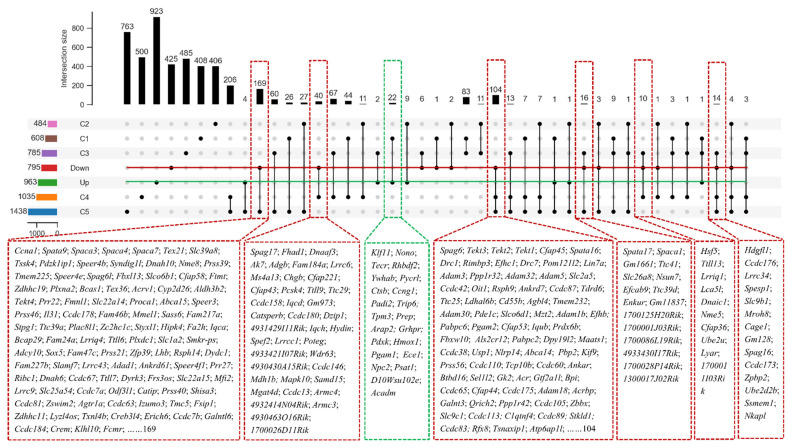
Enrichment of overlapping differentially expressed genes in specific highly expressed genes in male germ cells of various types of mice. C1, A1 to B spermatogonial cells and early G1 to S stage spermatogonial cells; C2, middle S stage and late S stage prolotene spermatocytes; C3, leptotene and zygotene spermatocytes; C4, pachytene spermatocyte; C5, diploid spermatocytes, MI and MII spermatocytes, as well as steps 1–2 spermatids; Down, down-regulated differentially expressed genes; Up, up-regulation of differentially expressed genes.

**Table 1 cells-11-01277-t001:** Statistical analysis of breeding assay of *Hnrnpk* cKO males.

Group	Number	Litters/Male (Mean ± SEM)	Pups/Litter (Mean ± SEM)
WT Control	6	3.67 ± 0.52	8.01 ± 1.17
*Hnrnpk* cKO	6	0	0

## Data Availability

The scRNA-seq data are publicly available and were deposited in the online supplementary material (https://www.nature.com/articles/s41422-018-0074-y) accessed on 3 April 2022.

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
