# Peer review of "Deletion of Hnrnpk Gene Causes Infertility in Male Mice by Disrupting Spermatogenesis"

_cells, 2022, doi:10.3390/cells11081277_

Round 1
Reviewer 1 Report
The authors have managed to specifically deplet the hnRNPK protein in germ cells, which has allowed them to study its role on spermatogenesis. However, there are some missing points, especially those related with the levels of protein expression in each cell, since authors have not quantified them. Moreover, the images from immunohystochemistry do not show the specific location of the protein as a result of the poor-quality acquisition. Before publishing this paper, the authors should address these crucial points and others herein detailed:
File 32: into spermatozoa (not one)
Figure 1: why in 1C) the Identification of hnRNPK cKO mice is not written as KO?
Materials and Methods
Overall, authors should indicate in each figure the number of biological replicates they have used. Moreover, for the immunohistochemistry and western blot, 3 is a very limited number of samples…
Gene expression: at least two housekeeping genes must be used, so one is missing. How much cDNA are authors using for the amplifications and what are the amplification conditions? Moreover, ΔΔCt can be used to represent the expression but using it for the statistics is not correct since this formula assumes that your control value is 1. Authors must use ΔCt instead.
WB: how much protein are the authors using to perform the WB?
TUNEL: authors should describe deeper the protocol? Have they performed a chromatin decondensation? Have the authors used the DNAse treatment as positive control?
Statistical assay: the normality of the samples must be checked, if it has been checked already, they should write it down, otherwise all the data must be re-evaluated to see if ANOVA or t-test can be applied.
Results and discussion
In figure 2: why have authors used YH2AX? It is a protein involved in the detection of double strand DNA damage, so it is not clear why they have selected it instead of acrosome staining, for example. Moreover, the images in D are of in such poor quality that it is not possible to distinguish anything. Authors must use a confocal microscope to improve the quality (not using that much the digital zoom, but a larger objective instead) and define better the location of this protein. The same for figure 4 and 5.
In line 219 they have written “relatively low levels” and “expression in the cytoplasm increased gradually in metaphase I and II” (among other sentences). However, this is all speculative since authors do not know it, unless they quantify the protein levels from the immune, which I highly recommend since authors have written an important part of the discussion about the expression levels of this protein in the different testicular cell types (lines 479-499)
All the description about mutant generation in line 3.2. should be explained before, in materials and methods. So, it has also more sense to describe first figure 1 than figure 2 and it’s also better for the reader’s understanding.
In line 278, the authors claim that they have “…examined the existence of the germ cells and somatic cells by H&E staining”. But regarding this point, they have just put a representative (I guess) picture of control and Hnrnpk cKO testes. To accurately evaluate that, a quantification of each cell type must be done (see 10.1016/j.envpol.2019.01.127). Furthermore, to evaluate the levels of Hnrnpk in the different cell type of the testicles, a quantification of the protein in the sections is needed. The western blot has the limitation of only providing an overall level of the protein the whole testicle.
As for the lumen diameter in supplementary figure 2E, it seems that the authors have assessed a different section, more longitudinal than the others that are transversal…
In figure 6, could authors add another Venn Diagram of DE protein-coding genes of the main testis cell types?
Line 477: meiosis genes
Line 493: is THE highest…
Author Response
Dear Reviewer,
Thank you very much for your attention and the referee’s evaluation and comments on our manuscript (Manuscript ID cells-1633546). We revised the manuscript in accordance with your comments, and carefully revised the manuscript to minimize typographical and grammatical errors and repetition rate. All changes made to the text are in red so that they may be easily identified. With regard to the your comments and suggestions, we wish to reply as follows:
File 32: into spermatozoa (not one)
Response: Thanks! We have revised them in the revised manuscript.
Figure 1: why in 1C) the Identification of hnRNPK cKO mice is not written as KO?
Response: Different primers were used to identify Hnrnpk loxP and Stra8-Cre mice genotype. Figure 1S shows the genotype identification of Hnrnpk loxP mice.
Materials and Methods
Overall, authors should indicate in each figure the number of biological replicates they have used. Moreover, for the immunohistochemistry and western blot, 3 is a very limited number of samples…
Response: Thanks! The number of biological replicates has been described in the text or in the figure notes in the revised manuscript. In general, 3 samples are sufficient to meet the basic requirements of WB and immunohistochemistry analysis.
Gene expression: at least two housekeeping genes must be used, so one is missing. How much cDNA are authors using for the amplifications and what are the amplification conditions? Moreover, ΔΔCt can be used to represent the expression but using it for the statistics is not correct since this formula assumes that your control value is 1. Authors must use ΔCt instead.
Response: Thanks! We have added the other housekeeping gene 18S and re-analyzed the spatio-temporal expression of Hnrnpk by qRT-PCR. We used 1.0 µg of RNA in reverse transcription. 0.4 µl 5-fold diluted cDNA was added in each qRT-PCR reaction system (total volume 20.0 µl). QRT-PCR reactions were performed using 0.4 µl of 5 x diluted cDNA, 0.2 µl primers (10 nM), 10.0 µl qRT-PCR mixture (FastStart Essential DNA Green Master), and 9.2 µl deionized water ina reaction volume totaling 20 µl. The qRT-PCR protocol was 1 cycle (50 °C for 2 min, 95 °C for 10 min) followed by 40 cycles of a two-stage temperature profile (95°C for 15 s, 60°C for 60 s). ΔCt is the Ct value of the target gene minus the Ct value of the steward gene, which is the corrected Ct value of the target gene. ΔΔCt is the ΔCt of the target gene minus the ΔCt of the largest of all the target genes. 2^(-ΔΔCt) reprents the relative expression value of the target gene. This is the most widely used method to calculate the relative expression values of target genes in qRT-PCR analysis. Not directly related to intergroup analysis.
WB: how much protein are the authors using to perform the WB?
Response: 30 µg protein per sample was loaded in each lane for SDS-PAGE. We have added it in the revised manuscript.
TUNEL: authors should describe deeper the protocol? Have they performed a chromatin decondensation? Have the authors used the DNAse treatment as positive control?
Response: Thanks! We have added more detailed protocol of TUNEL assay and Meiotic chromosomal spreading in the revised manuscript. There was no positive control experiment for the DNAse treatment in TUNEL assay.
Statistical assay: the normality of the samples must be checked, if it has been checked already, they should write it down, otherwise all the data must be re-evaluated to see if ANOVA or t-test can be applied.
Response: Yes, it has been checked already. We have revised them in the revised manuscript.
Results and discussion
In figure 2: why have authors used YH2AX? It is a protein involved in the detection of double strand DNA damage, so it is not clear why they have selected it instead of acrosome staining, for example. Moreover, the images in D are of in such poor quality that it is not possible to distinguish anything. Authors must use a confocal microscope to improve the quality (not using that much the digital zoom, but a larger objective instead) and define better the location of this protein. The same for figure 4 and 5.
Response: Thanks! Conveniently, phosphorylation on the C-terminus of histone H2A.X (γH2A.X) is a well-known indicator of the presence of DSBs and the ensuing DNA damage response. DSBs and DSB repair are the key events in spermatogenesis meiosis. Thus, γH2A.X is also used to assess meiotic progression, and has a role in condensing and inactivating sex chromosomes in male meiosis in mice (Fernandez-Capetillo et al, Dev. Cell, 2002, 4:497-508). The specific antibodies against γH2A.X have been used to represent the germ cell in meiosis and distinguish the different spermatocyte types during meiosis (leptotene, zygotene, pachytene and diplotene). The quality of Figure 2D is caused by local enlargement of Figure 1C, which does not affect the presentation of results. The confocal microscope is a relatively expensive instrument, and there is no relevant instrument in our laboratory at present. Although better image quality can be obtained through confocal microscope, we think that images taken by ordinary fluorescence microscopy can also meet basic needs, especially for images based on qualitative analysis.
In line 219 they have written “relatively low levels” and “expression in the cytoplasm increased gradually in metaphase I and II” (among other sentences). However, this is all speculative since authors do not know it, unless they quantify the protein levels from the immune, which I highly recommend since authors have written an important part of the discussion about the expression levels of this protein in the different testicular cell types (lines 479-499).
Response: Thanks! Indeed, immunofluorescence staining is mainly used for qualitative analysis. Cannot be used for quantitative comparison of different images. However, the fluorescence intensity from the same image can be compared because the background is the same. In this study, when we analyzed the expression of hnRNPK in different types of germ cells, we compared cells from the same section. Thus, the fluorescence intensity of hnRNPK in different cells can be regarded as the strength of hnRNPK expression. In addition, we have also revised the relevant descriptions in the revised manuscript.
All the description about mutant generation in line 3.2. should be explained before, in materials and methods. So, it has also more sense to describe first figure 1 than figure 2 and it’s also better for the reader’s understanding.
Response: Thanks! We have changed Figure 1 to Figure 2 to appear in the revised manuscript.
In line 278, the authors claim that they have “…examined the existence of the germ cells and somatic cells by H&E staining”. But regarding this point, they have just put a representative (I guess) picture of control and Hnrnpk cKO testes. To accurately evaluate that, a quantification of each cell type must be done (see 10.1016/j.envpol.2019.01.127). Furthermore, to evaluate the levels of Hnrnpk in the different cell type of the testicles, a quantification of the protein in the sections is needed. The western blot has the limitation of only providing an overall level of the protein the whole testicle.
Response: Thanks! Through immunofluorescence, different types of cells in testis of WT control and Hnrnpk cKO mice have been detected, and statistical analysis has been conducted too, as shown in Figure 4. In addition, immunofluorescence staining cannot be used for quantitative analysis, but can only be used for qualitative analysis. Due to the numerous cell types in testis, it is difficult to separate one from the other under current research conditions. Therefore, it is difficult to accurately quantify the expression level of a protein in a particular cell. It is acceptable to judge the expression of a protein in different cell types in the same testicular section by the intensity of immunofluorescence staining.
As for the lumen diameter in supplementary figure 2E, it seems that the authors have assessed a different section, more longitudinal than the others that are transversal…
Response: Some oval cavities in supplementary figure 2E are mainly caused by the malposition of paraffin-embedded epididymis samples. In the analysis, we mainly counted the diameter of the circular cavity. In fact, with epididymis sections we're more concerned with the presence of clusters of sperm.
In figure 6, could authors add another Venn Diagram of DE protein-coding genes of the main testis cell types?
Response: There are so many cell types in the testes that it can be difficult to distinguish which cell type these differentially expressed genes belong to.
Line 477: meiosis genes
Response: We have revised it in the revised manuscript.
Line 493: is THE highest…
Response: Thanks! We have revised it in the revised manuscript.
Reviewer 2 Report
Xu and colleagues present work investigating the role of Hnrnpk in spermatogenesis. They create a floxed Hnrnpk allele which is then crossed to a Stra8-cre line in order to selectively delete in the male germline, finding that Hnrnpk is required for male germ cell development and fertility. A series of immunostaining experiments reveal that spermatogenesis fails at the pachytene stage and RNA-seq shows that meiosis is unable to proceed. For the most part this is a strong manuscript, but I have some concerns that should be addressed before considering publication.
Major concerns:
- Fig 2a – BioGPS lists the expression of Gapdh as very low in testis, therefore the pattern of Hnrnpk expression presented can be explained by normalising to Gapdh, rather than Hnrnpk being highly expressed in this tissue. Authors need to use multiple house-keepers for noramlisation of data in Fig 2a,b. Also, please define what ‘a,b,c, a1, a2, d,e’ on the graphs represent.
- Please provide quantification of data for Fig 3a,b and statistical testing.
- More should be done to show that Hnrnpk is deleted at the RNA/protein level. Please provide qRT-PCR in the WT and CKO. Also, the Western shown in Fig 3e is not striking– this should be quantified and normalised to the control so readers can assess the subtle difference. Also, the remaining Hnrnpk observed in the CKO Western could be due to partial penetrance of the conditional deletion. The authors should assess the penetrance of the deletion.
- The data presented in Fig 5a,b should be scored and quantified, so readers can assess how many cells fell into each category.
- Fig 5d is hard to interpret and should be shown as two separate pie charts, one each for the UP and Down genes.
- One of the conclusions of the manuscript is that Hnrnpk is acting as a transcription factor (line 44), however you can’t tell this from the RNAseq. All that can be deduced from the RNA-seq is that the transcriptome changes without Hnrnpk, but nothing can be inferred about the mechanism. Hnrnpk could be acting as a transcription factor, or could be post-transcriptional regulation, or secondary effects. The authors need to change their conclusion here. Also, please cite papers and discuss the evidence from the literature for Hnrnpk being a transcription factor.
- The discussion was hard to read and a bit unstructured. The manuscript would be greatly improved by reworking the discussion.
- There is very little statistical information in the manuscript, including what tests were performed, number of replicates etc. This information should be included. Also, the manuscript relies heavily on bar graphs with error bars. The manuscript would benefit from the graphs displaying each individual data point, so that variance and replicate number are easily apparent from the graph.
Minor concerns:
Sentence on lines 55-58 needs citations.
Figure 2 is cited before Fig 1 in the text. Change the order of the figures.
Line 198, Define BAT and iWAT.
Line 208, explain why you stained for γ-H2A.X
Fig2e not referred to in text.
Line 256 – 263, writing very hard to understand.
Error bars are missing on Supp Fig 3a,c. Should really plot individual data points rather than error bars though.
Line 504, the citation 38 appears wrong here.
Author Response
Dear Reviewer,
Thank you very much for your attention and the referee’s evaluation and comments on our manuscript (Manuscript ID cells-1633546). We revised the manuscript in accordance with your comments, and carefully revised the manuscript to minimize typographical and grammatical errors and repetition rate. All changes made to the text are in red so that they may be easily identified. With regard to the your comments and suggestions, we wish to reply as follows:
Major concerns:
- Fig 2a – BioGPS lists the expression of Gapdh as very low in testis, therefore the pattern of Hnrnpk expression presented can be explained by normalising to Gapdh, rather than Hnrnpk being highly expressed in this tissue. Authors need to use multiple house-keepers for noramlisation of data in Fig 2a,b. Also, please define what ‘a,b,c, a1, a2, d,e’ on the graphs represent.
Response: Thanks! We have added the other housekeeping gene 18S and re-analyzed the spatio-temporal expression of Hnrnpk by qRT-PCR. The difference between different letters is significant (P < 0.05), while the difference between the same letters is not significant (P > 0.05); n = 3. We have defined it in Figure 1A and B in the revised manuscript.
- Please provide quantification of data for Fig 3a,b and statistical testing.
Response: We have added the weight of mice (mean ± SEM) of Figure 3A in the revised manuscript. The data of Figure 3A is displayed in Figure 3C.
- More should be done to show that Hnrnpk is deleted at the RNA/protein level. Please provide qRT-PCR in the WT and CKO. Also, the Western shown in Fig 3e is not striking– this should be quantified and normalised to the control so readers can assess the subtle difference. Also, the remaining Hnrnpk observed in the CKO Western could be due to partial penetrance of the conditional deletion. The authors should assess the penetrance of the deletion.
Response: In this study, only the sequence of the Hnrnpk gene from the third intron to the seventh intron is knocked out by loxP-cre system. It destroys the initiation of translation of Hnrnpk transcript and cannot produce hnRNPK protein, thus silencing hnRNPK. In fact, the expression of Hnrnpk at the transcriptional level has not changed, so it is meaningless to detect changes at the transcriptional level. In addition to germ cells, hnRNPK protein is also expressed in sertoli cells, stromal cells, myoid cells, fibroblasts and other somatic cells, while Sre8-cre mediated conditional knockout only starts from type A spermatogonial cells. It is normal to detect weak expression of hnRNPK in the testes of cKO mice. In addition, immunofluorescence staining results of hnRNPK can also explain the deletion of hnRNPK in germ cells.
- The data presented in Fig 5a, b should be scored and quantified, so readers can assess how many cells fell into each category.
Response: Thanks! This is a good suggestion. The scoring and quantification of chromosome spread results will be better to understand the effects of hnRNPK deficiency on meiosis I processes. However, considering that there was a small amount of off-target phenomenon in the Stra8-Cre mediated hnRNPK knockout in germ cell, immunofluorescence and chromosome spread results showed that there were a small number of hnRNPK positive germ cells in the testis of hnRNPK cKO mice. This makes it difficult to analyze the immunofluorescence and chromosome spread of Hnrnpk cKO mice, and it is also difficult to scored and quantified of chromosome spread results exactly. In order to obtain reliable results, we used more biological duplications (n = 6) in chromosome spreading and used hnRNPK and SYCP3 co-staining to better distinguish hnRNPK positive and hnRNPK negative spermatocytes.
- Fig 5d is hard to interpret and should be shown as two separate pie charts, one each for the UP and Down genes.
Response: If we use two figures, the layout of the whole diagram will be difficult. In fact, overlapping and non-overlapping genes can be distinguished nicely from the v-plots of the four groups.
- One of the conclusions of the manuscript is that Hnrnpk is acting as a transcription factor (line 44), however you can’t tell this from the RNAseq. All that can be deduced from the RNA-seq is that the transcriptome changes without Hnrnpk, but nothing can be inferred about the mechanism. Hnrnpk could be acting as a transcription factor, or could be post-transcriptional regulation, or secondary effects. The authors need to change their conclusion here. Also, please cite papers and discuss the evidence from the literature for Hnrnpk being a transcription factor.
Response: Thanks! In the past, many studies have shown that K can play a role as a transcription factor in many biological processes (reference13, 15, 27, 29). In germ cells, hnRNPK is mainly expressed in the nucleus, and the results of chromosome spreading also show that hnRNPK is dotted along chromosomes. The deletion of hnRNPK also resulted in significant changes in the transcription levels of a large number of genes. These results can preliminarily indicate that hnRNPK may play a more regulatory role as a transcription factor in spermatogenesis at the transcription level. Of course, post-transcriptional regulation cannot be ruled out, and more experiments such as ChIP or EMSA are needed to further reveal the molecular mechanism of hnRNPK's regulation of spermatogenesis. In addition, the description of the conclusions has been appropriately revised in the revised manuscript.
- The discussion was hard to read and a bit unstructured. The manuscript would be greatly improved by reworking the discussion.
Response: Thanks! We have revised the discussion section appropriately in the revised manuscript.
- There is very little statistical information in the manuscript, including what tests were performed, number of replicates etc. This information should be included. Also, the manuscript relies heavily on bar graphs with error bars. The manuscript would benefit from the graphs displaying each individual data point, so that variance and replicate number are easily apparent from the graph.
Response: Thanks! We have added the related statistical information in the revised manuscript.
Minor concerns:
Sentence on lines 55-58 needs citations.
Response: We have added the citation in the revised manuscript.
Figure 2 is cited before Fig 1 in the text. Change the order of the figures.
Response: Thanks! We have changed Figure 1 to Figure 2 to appear in the revised manuscript.
Line 198, Define BAT and iWAT.
Response: We have defined them in Figure 2 in the revised manuscript.
Line 208, explain why you stained for γ-H2A.X
Response: Conveniently, phosphorylation on the C-terminus of histone H2A.X (γH2A.X) is a well-known indicator of the presence of DSBs and the ensuing DNA damage response. DSBs and DSB repair are the key events in spermatogenesis meiosis. Thus, γH2A.X is also used to assess meiotic progression, and has a role in condensing and inactivating sex chromosomes in male meiosis in mice (Fernandez-Capetillo et al, Dev. Cell, 2002, 4:497-508). The specific antibodies against γH2AX has been used to represent the germ cell in meiosis and distinguish the different spermatocyte types during meiosis (leptotene, zygotene, pachytene and diplotene).
Fig2e not referred to in text.
Response: We have referred it in the revised manuscript.
Line 256 – 263, writing very hard to understand.
Response: We have revised this sentence in the revised manuscript.
Error bars are missing on Supp Fig 3a,c. Should really plot individual data points rather than error bars though.
Response: The histograms in Figure 3A and 3C represent the log2(Fold change) in RNA-Seq and qRT-PCR analysis between the two groups. Thus, there is no error bar value.
Line 504, the citation 38 appears wrong here.
Response: Thanks! We have inserted a wrong reference. We have changed it to the correct one in the revised manuscript.
Round 2
Reviewer 1 Report
I appreciate the author’s answer and the improvements they have made. However, some points must be addressed so that the data fit better with the discussion and conclusions they present.
How can it be that, in the first version, the expression of hnRNPK in the brain (and other tissues) was so different if authors have not added any more replicates? If these changes are only due to housekeeping genes, it seems that they’re not constitutive expressed. Can the authors show the data regarding the expression of housekeeping genes for figure 1A?
Regarding the statistical analysis, if authors have analyzed the normality of the samples, why they have not written it down in the statistical part of the methods? I’m pretty sure that not all the data they have analyzed are normal (especially those presented in table 1), so analyzed the difference among groups using t-test o r ANOVA is not correct.
As far as immunofluorescence is concerned, I completely disagree with the authors. Of course, it can be used as a quantitative method, if the settings of the microscopy are the same for every image taken as well as the conditions for the . I’ve suggested to quantify the levels of hnRNPK since it is crucial for the understanding of the protein’s role in each subpopulation of testicular cells.
In supplementary figure 2E. If there was a problem in the orientation of the organs, authors should repeat it for these sample. Otherwise, this measure makes no sense since it cannot be compared to the control lumen diameter.
Author Response
Dear Reviewes,
Thanks again for your comments on our manuscript (Manuscript ID cells-1633546). We revised the manuscript in accordance with the your comments, and carefully revised the manuscript to minimize typographical and grammatical errors. All changes made to the text are in red so that they may be easily identified. With regard to the your comments and suggestions, we wish to reply as follows:
1. How can it be that, in the first version, the expression of hnRNPK in the brain (and other tissues) was so different if authors have not added any more replicates? If these changes are only due to housekeeping genes, it seems that they’re not constitutive expressed. Can the authors show the data regarding the expression of housekeeping genes for figure 1A?
Response: Thanks a lot. Housekeeping genes are constitutively expressed in all tissues to maintain cellular functions, which were usually used to represent appropriate controls for qRT-PCR and northern blot when comparing the expression levels of genes in several tissues. However, the expression level of the housekeeping genes may vary among tissues or cells and may change under certain circumstances. There is no absolutely stable expression of housekeeping genes among different tissues. In theory, the more housekeeping gene you choose, the more reliable the expression value will be. The difference between the two tissue expression profiles of Hnrnpk may be mainly due to the stability of the housekeeping gene. We searched the expression list of Gapdh in the NCBI Database and also found that Gapdh was different in different tissues, with relatively low expression in testis and relatively high expression in brain. With the introduction of the second housekeeping gene, the differences will be appropriately eliminated, which also increases the reliability of Hnrnpk expression. The result shown in Figure 1A is the relative expression value of Hnrnpk calculated by 2^(-ΔΔCt) method. In this calculation method, the CT value of the housekeeping gene is used for correction, so the expression of housekeeping gene cannot be shown in the Figure 1A.
2. Regarding the statistical analysis, if authors have analyzed the normality of the samples, why they have not written it down in the statistical part of the methods? I’m pretty sure that not all the data they have analyzed are normal (especially those presented in table 1), so analyzed the difference among groups using t-test o r ANOVA is not correct.
Response: Thanks! The two sets of data in Table 1 are actually the difference between "yes" and "no", and we did not conduct t-test analysis. We only counted the Litters/male and Pups/litter. This was enough to indicate that the mice were sterile.
As far as immunofluorescence is concerned, I completely disagree with the authors. Of course, it can be used as a quantitative method, if the settings of the microscopy are the same for every image taken as well as the conditions for the. I’ve suggested to quantify the levels of hnRNPK since it is crucial for the understanding of the protein’s role in each subpopulation of testicular cells.
3. Response: Thanks! It is true that immunofluorescence cannot be used for quantitative analysis of proteins. At present, there is no good method to quantify the fluorescence intensity of immunofluorescence staining, even under the same fluorescence background. In this study, we only used immunofluorescence to analyze the expression trend of H during the development of male germ cells. In addition, the description of fluorescence intensity has also been revised and replaced with descriptions such as ‘weak’ or ‘strong’ in the revised manuscript. In fact, many researches have also used immunofluorescence to analyze protein expression in different germ cell types during spermatogenesis (Bao et al., Plos Genetics, 2014, 12: e1004825; Dong et al., Nature Communications, 2019, 10:4705; Liu et al., Development , 2021, 19: dev198374; Wu et al., Front. Cell Dev. Biol., 2021, 9:715733; Wang et al., Scitienfic Report, 2015, 5:11031; Wang et al., Development, 2021, 148, dev196295).
4. In supplementary figure 2E. If there was a problem in the orientation of the organs, authors should repeat it for these sample. Otherwise, this measure makes no sense since it cannot be compared to the control lumen diameter.
Response: Thanks! The H&E staining of epididymis section is mainly to see whether there are clusters of sperm, and the diameter of epididymis lumen is irrelevant. We deleted the statistical analysis of epididymis lumen diameter in the revised manuscript.
Reviewer 2 Report
The revisions have adequately addressed my concerns. I have no further requests.
Author Response
Response: Thank you very much for your attention and comments on our manuscript.